psychology

communication, policy, evidence, acceptability, attitudes, beliefs

**Author for correspondence:**
T. M. Marteau
e-mail: tm388@medschl.cam.ac.uk

†Present address: Institute of Public Health, Robinson Way, University of Cambridge, Cambridge CB2 0SR.

# Communicating the effectiveness and ineffectiveness of government policies and their impact on public support: a systematic review with meta-analysis

J. P. Reynolds[1], K. Stautz[1], M. Pilling[1],
S. van der Linden[2] and T. M. Marteau[1,†]

[1]Behaviour and Health Research Unit, University of Cambridge, Cambridge, UK
[2]Department of Psychology, University of Cambridge, Cambridge, UK

 JPR, 0000-0003-1536-1557; MP, 0000-0002-7446-6597;
TMM, 0000-0003-3025-1129

Low public support for government interventions in health, environment and other policy domains can be a barrier to implementation. Communicating evidence of policy effectiveness has been used to influence attitudes towards policies, with mixed results. This review provides the first systematic synthesis of such studies. Eligible studies were randomized controlled experiments that included an intervention group that provided evidence of a policy's effectiveness or ineffectiveness at achieving a salient outcome, and measured policy support. From 6498 abstracts examined, there were 45 effect sizes from 36 eligible studies. In total, 35 ($N = 30\,858$) communicated evidence of effectiveness, and 10 ($N = 5078$) communicated evidence of ineffectiveness. Random effects meta-analysis revealed that communicating evidence of a policy's effectiveness increased support for the policy (SMD = 0.11, 95% CI [0.07, 0.15], $p < 0.0001$), equivalent to support increasing from 50% to 54% (95% CI [53%, 56%]). Communicating evidence of ineffectiveness decreased policy support (SMD = −0.14, 95% CI [−0.22, −0.06], $p < 0.001$), equivalent to support decreasing from 50% to 44% (95% CI [41%, 47%]). These findings suggest that public support for policies in a range of domains is sensitive to evidence of their effectiveness, as well as their ineffectiveness.

# 1. Introduction

Obesity rates are high and rising worldwide [1], human-caused climate change proceeds towards a level incompatible with life in a growing number of regions [2], and mass shootings occur almost daily in the USA [3]. These problems, among many others, require the action of policy makers, yet low public support can present a barrier for such action, especially in an increasingly complex and politicized information environment [4–7]. There have been many attempts to increase support for various policies across issue domains, and communicating evidence of a policy's effectiveness has potential as one possible strategy [8]. The current study aims to systematically synthesize the evidence for changing public attitudes and support for policies by communicating evidence about a policy's effectiveness at achieving its goal. This is the first study of which we are aware to conduct such a synthesis.

The perceived effectiveness of a policy at achieving its goal has consistently been found as one of the strongest predictors of support over a range of policies [8–14]. Reflecting these findings, a number of studies have communicated evidence that a policy is effective as an intervention to increase policy support, with mixed results. Studies that have communicated unconfounded evidence that a policy is effective have mostly found increased support for the policy, with perceived effectiveness acting as the mechanism behind this change [8,15–17]. Studies that have communicated messages containing evidence of effectiveness in addition to further information have yielded mixed results, with the mechanism(s) remaining unclear [18–21]. Further studies communicating evidence that a policy is ineffective or that it has undesirable outcomes, such as economic and health costs, have also led to mixed results [15,22,23]. Even within a single study, the same intervention has been shown to have different effects on different policies [18], leading to uncertainty about the effectiveness of such approaches.

This work can be understood more broadly in terms of how changes in relevant beliefs (e.g. the perceived effectiveness of a policy) can engender a change in basic attitudes (e.g. support for the policy; [24]). Public support can be defined as how individuals feel and think about the implementation of a policy. As such, this area of research sits within literatures in judgement and decision-making concerning the impact of information—including misinformation—on a broad range of psychological and behavioural outcomes, including voting and support for policies. Importantly, in a so-called post-truth society where the nature of evidence is increasingly contested [25], recent studies have been stimulated by a concern that providing factual information has little impact on people's beliefs and might sometimes even have the opposite of the intended effect, serving only to entrench pre-existing beliefs when these differ markedly from the evidence being presented. In general, there are three possible responses to being presented with evidence: maintenance of one's current beliefs, polarization of an existing belief away from beliefs consonant with the evidence, and updating of beliefs in a direction consistent with the evidence. Although people are generally motivated to hold accurate beliefs about the world [26], directionally biased assimilation can occur when people selectively credit or discredit 'evidence' to arrive at a preferred conclusion. The confirmation bias—the seeking and interpretation of evidence that is consistent with one's intial beliefs—has emerged as a central mechanism for understanding why people do not change their beliefs when provided with evidence [27]. In addition, sometimes people update their beliefs in the opposite direction of the evidence. For example, one influential study found that, when considering the effectiveness of capital punishment on crime prevention, exposure to mixed evidence caused students to strengthen their prior convictions [28]. This process is variously described as 'belief polarization', 'boomerang' or 'backfire effect' [29]. Subsequent studies, however, found that true belief polarization is not the norm, and in fact, a relatively rare phenomenon; 'by and large, citizens heed factual information, even when such information challenges their ideological commitments' [30–32].

Indeed, the most likely response when confronted with disconfirming or contrary evidence is for the individual to move their belief toward the evidence [33–36]. This is often observed as a mean change on a response scale. This indicates that while people may report different numbers on a scale, they may not have shifted their beliefs categorically from one side to another, for example from disbelieving to believing in the existence of human-caused climate change. The number of people who change their position entirely is smaller than the number who alter specific beliefs [37,38]. At the same time, evidence-based opinion formation is hampered by the fact that in the current media environment, falsehoods can spread deeper and faster than factual information [39]. In addition, the advent of

social media has enabled communities of like-minded people to easily share ideas that conform with and reinforce existing beliefs (i.e. echo chambers; [40,41]). One common example of echo-chambers are those organized around politics [42]. It is unsurprising then, that some beliefs and attitudes are harder to change than others, such as those that have strong political implications [29,35,43]. Attitudes towards the implementation of policies are one such area that may be harder to shift, especially given that public support for some policies is increasingly influenced by political identities [44], for example, around climate change, abortion, and gun control. Yet, even so, a large literature in persuasion psychology shows that the framing of information can significantly alter the value and weight people attach to their beliefs about policies and their effectiveness [45].

In other words, different ways of communicating the same information may be more or less successful at changing attitudes [46] and therefore need to be considered. For example, quantitative estimates of policy effectiveness are potentially more effective than qualitative estimates in allowing people to understand gradations of effectiveness [47,48]; e.g. the difference between very effective and extremely effective is unclear whereas the difference between 10% reduction in crime and 15% reduction in crime is clear. Despite this potential benefit in comprehension from quantifying effectiveness, it is not clear if this would translate into greater support for that policy. The use of uncertainty qualifiers has seen a surge of interest in recent years [49,50]. However, it remains unclear how the use of uncertainty qualifiers when describing policy effectiveness would affect attitudes towards the policy. Because people are generally averse to ambiguity [51], if the evidence about the effectiveness of the policy were presented as uncertain, this may mute any change in attitudes towards the implementation of that policy.

Using meta-analytic evidence synthesis, the aim was to assess whether communicating evidence that a policy is effective or ineffective changes support for the policy and if so, by how much. The authors hypothesize that both sets of evidence will change people's attitudes in the direction consistent with the evidence that they receive. Additionally, it is predicted that the effects will be larger when communicating ineffectiveness information due to negativity biases [52]. The moderating effects of policy domain and intervention characteristics were also tested.

# 2. Method

This systematic review is reported in line with PRISMA (preferred reporting items for systematic reviews and meta-analyses) guidelines [53]. The review protocol was prospectively registered in the PROSPERO database (ID: http://www.crd.york.ac.uk/PROSPERO/display_record.php?ID=CRD42017079524). The data and code are available at https://osf.io/4gjur/.

## 2.1. Eligibility criteria

Policies eligible for inclusion were actual or hypothetical policy interventions to tackle a problem that might be implemented by local or national governments, or by supranational bodies, such as the World Health Organization. Eligible studies include randomized experiments in which one group of participants received information about the (in)effectiveness or impact of at least one policy and a control group of participants that did not receive any information about the (in)effectiveness or impact of the policy. Within-participants, between-participants and quasi-experimental (i.e. experiments without random assignment) designs were also eligible.

There were no restrictions on types of participants. Eligible interventions for the evidence of effectiveness analysis were those that provided information regarding evidence of the impact of the policy, in terms of the potential benefits, or how the policy would address a specified problem (i.e. be effective). This includes evidence that discusses qualitative or quantitative descriptions of the magnitude of the effect, or assertions that describe the effect without referring to its magnitude.

Eligible interventions for the evidence of ineffectiveness analysis were those that provided information regarding evidence of the impact of the policy, in terms of the potential harms, or how the policy would fail to achieve its intended purposes (i.e. be ineffective). This includes evidence that discusses qualitative or quantitative descriptions of the magnitude of the effect, or describe the effect without referring to its magnitude. Both effectiveness and ineffectiveness interventions were eligible if they asserted that a policy was effective, even if they did not specify that the information originated from an expert or from research, e.g. 'this tax would lead to a 1.6% reduction in obesity' [8]. Interventions were ineligible if the information was presented as coming from the general public or

non-experts, e.g. 'Some people believe that increasing trade with other nations creates jobs and allows you to buy goods and services at lower prices' [54].

Eligible comparators were control groups that received no information pertaining to the effectiveness or impact of the policy. Eligible outcomes were the acceptability of a specific policy or set of policies, defined as the level of support or attitude toward the implementation of policy/policies, measured using response scales that allow for a binary assessment (i.e. support or oppose), or a graded degree of support or opposition. Measures of support for public or societal action in general, not linked to a specific action or policy, were excluded [35].

## 2.2. Literature search

The search strategy was developed with the assistance of an information scientist. Nine electronic databases were searched: ASSIA, EconLit, EMBASE, Open Grey, PsycINFO, Public Affairs Information Service, PubMED, Science Direct, and Web of Science. There were no restrictions on language or publication date. Search terms were as follows:

(accept* OR support* OR favour* OR agree* OR attitude* OR opinion* OR perspective*) AND (policy OR regulation OR intervention OR proposal OR action OR government) AND experiment AND (information OR vignette OR narrative OR evidence OR statement OR frame OR framing) AND (effect* OR impact OR outcome OR consequence)

These terms were selected and developed based on terms that were used in the eligible papers that we had already located prior to the review. Database searches were completed up to October 2017. Two researchers (K.S. and J.P.R.) independently screened title-abstract records for eligibility. Screening discrepancies were resolved by discussion. Full texts of provisionally eligible records were retrieved via electronic library resources, screened independently by two researchers (K.S. and J.P.R.), and judged to be eligible or excluded with reasons recorded. Database searches were supplemented with snowball searches and forward citation tracking (using Google Scholar) of eligible articles, and reference list searches of relevant review articles. Requests for further unpublished data were made to the corresponding authors of eligible articles.

## 2.3. Data extraction

### 2.3.1. Information extracted

One researcher (either K.S. or J.P.R.) extracted information from eligible studies. Once extraction was complete, each researcher cross-checked the extraction completed by the other. Discrepancies in extracted information were resolved by discussion and by consulting with a third or fourth researcher if necessary (T.M.M. or S.v.d.L.).

The following information was extracted from each study: author name, year of publication, country in which data were collected, setting, study design, sample characteristics (age, gender, ethnicity, socio-economic position), inclusion/exclusion criteria for participants, policy domain and proposed policy, details of the intervention(s) (using the below coding scheme), details of the information given to the control group, dependent variable(s), method of analysis, outcome data, and information needed for quality assessment.

### 2.3.2. Coding

A coding scheme for interventions was developed prior to data extraction and conducted by two researchers in duplicate. Inter-rater agreement was high (93%). Discrepancies in extracted information were resolved by discussion and by consulting with a third or fourth researcher if necessary. Ten features of each intervention were coded: other information communicated (the nature of the problem being addressed by the policy, mechanism by which the policy affects the problem, defining/explaining concepts, inoculating against alternative policies, downsides to the proposed policy), content of message (quantitative statement of effect, qualitative statement of effect, assertion of effect; see electronic supplementary material, table S3 for examples), medium (e.g. text on screen, face to face interview), length of treatment, proportion of information that was a statement of effectiveness, readability (assessed using the Gunning-Fog Index), source of information used, attributed source of information, and stated level of uncertainty, if any.

### 2.3.3. Missing data

Where original studies did not provide adequate data for use in meta-analysis or intervention coding, requests for further information were made to the corresponding authors of those studies. Of 24 authors contacted, 17 provided these requested data. The full text of the experimental interventions was unavailable for only one of the papers included in the meta-analyses and was therefore not coded. In the effectiveness meta-analysis, effect sizes and associated confidence intervals (CIs) were calculated from means, standard deviations and sample sizes ($k = 30$), $t$ statistics and sample sizes ($k = 4$), and odds ratios and sample sizes ($k = 1$). This means that one of the 35 effect sizes in the evidence of effectiveness meta-analysis was converted from a dichotomous outcome ([55], formula 7.1). In one case [56], where only total sample size was available, it was assumed that the group sizes were equal. In the evidence of ineffectiveness meta-analysis, most effect sizes and associated confidence intervals were calculated using means, standard deviations and sample sizes ($k = 9$), and one was calculated using a $t$ statistic and sample size.

## 2.4. Risk of bias

The quality assessment tool for quantitative studies [57] was used to provide a methodological rating for each eligible study on the following categories: selection bias, study design, blinding, data collection methods, and withdrawals and dropouts. One category—confounders—was not factored into the quality score following advice from a reviewer. Ratings on these categories were then used to create a summary rating for the study: weak, moderate or strong. These ratings were conducted by two separate researchers, with disagreements resolved by discussion and a third researcher in certain cases. The agreement between the two primary reviewers was substantial (linear weighted $\kappa = 0.78$). Sensitivity analyses were conducted to determine if the main results remained after only including studies that were not at high risk of bias. Funnel plots and Egger's regression were used to detect small study bias (funnel plot asymmetry), in which smaller studies have larger effect sizes. This can indicate publication bias or other forms of bias. Where these funnel plots suggested that bias was present, the trim and fill method was used to produce the best estimate of the unbiased effect size [55,58]. This approach identifies any potential funnel plot asymmetry and imputes 'missing' studies which should remove the asymmetry.

## 2.5. Synthesis of results

Quantitative synthesis (meta-analysis) was used to calculate summary effect sizes. Our primary meta-analysis examined the effect of presenting information of effectiveness that was in favour of a policy, compared to no information, on public support. A secondary meta-analysis examined the effect on public support of presenting evidence of ineffectiveness and/or information that the policy had outcomes that were not in its favour when compared to no information.

As eligible studies used a range of different measures to assess public support, study-level standardized mean differences (SMDs; specifically, Hedges' $g$) were computed between comparison groups with 95% confidence intervals [55]. To ensure independence of observations in any meta-analysis, the following decision rules were followed prior to conducting the analyses: (i) in studies that included multiple eligible outcome measures, the combined means and variances were calculated using standard formulae ([55] equations 24.1 and 24.2); and (ii) when multiple interventions were eligible, the intervention containing evidence of effectiveness alone would be chosen over an intervention containing evidence of effectiveness *and* information on the nature of the problem. In cases where no intervention could be singled out using this method, the eligible intervention groups were combined into a single group as recommended by the Cochrane Handbook [58].

For each meta-analysis, random effects were used rather than fixed effects due to significant heterogeneity between studies. Statistical heterogeneity was assessed using the $\chi^2$ test and the $I^2$ statistic. As stated in the review registration, follow-up analyses were conducted by policy domain (health, environment, other), and intervention characteristics: the readability of the interventions (assessed with the Gunning Fog index), by the presence of uncertainty in the evidence of effectiveness, and by how effectiveness is presented (assertions that the policy has a particular outcome versus descriptions of the magnitude by which the policy has a particular outcome). Meta-regressions were conducted to investigate these potential moderation effects. Sensitivity analyses were also conducted to determine if the main results remained unaltered after excluding studies that were at high risk of bias

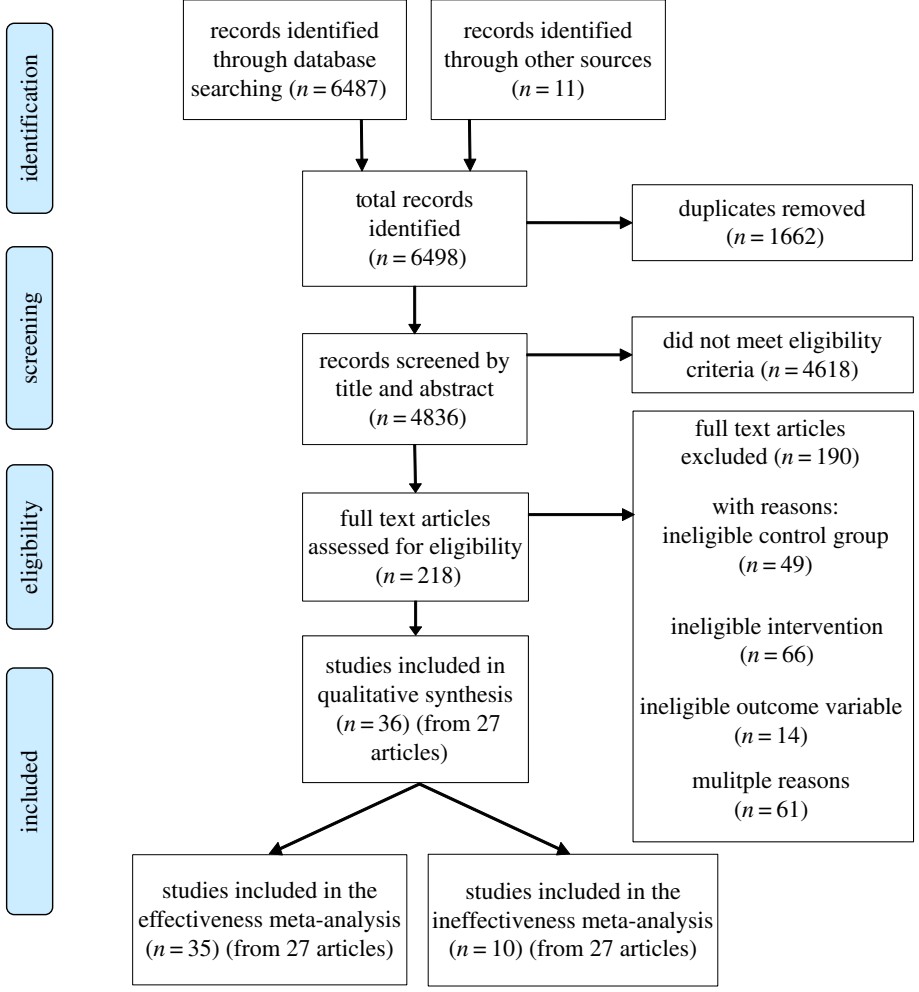

**Figure 1.** PRISMA flow chart displaying study flow.

and also after excluding studies that included confounded interventions, e.g. interventions that included evidence of effectiveness plus other information thought to influence policy support. R v. 3.3.3 package metafor were used to run the meta-analyses and meta-regressions [59,60].

The pooled effect sizes and associated confidence intervals were converted to changes in policy support proportions using a number needed to treat formula (table 2; [61]). The R code used to estimate changes in policy support proportions can be found in the supplement.

# 3. Results

## 3.1. Study selection

Figure 1 displays the flow of studies through the systematic review process. A total of 4836 study records were screened based on their titles and abstracts. Full-text screening of 218 articles that were judged to be potentially eligible resulted in 26 that met the inclusion criteria. One further eligible article was identified after contact with authors, resulting in 27 articles. Within these 27 articles, 35 studies were eligible for the effectiveness meta-analysis ($N = 30\,858$), 10 studies were eligible for the ineffectiveness meta-analysis ($N = 5078$), and one study—that communicated evidence of effectiveness—included insufficient information to be included in the meta-analysis [62].

## 3.2. Study characteristics

Of the 35 effect sizes included in the effectiveness meta-analysis, 20 were in the domain of health policies, nine in environmental policies, and six in other areas, including gun crime and education. Of the 10 effect

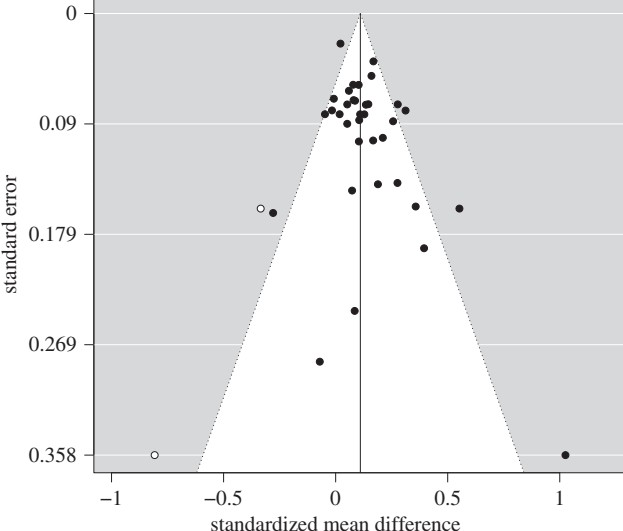

**Figure 2.** A funnel plot for the effectiveness meta-analysis with trim and fill.

sizes included in the ineffectiveness meta-analysis, six were in the domain of health policies, three in environmental policies, and one in another area—war. The majority of the studies were conducted in the USA (25/36) and despite the eligibility criteria being open to other designs, all included studies were randomized between-participants experiments. The control groups in most included studies (33/36) received no additional information beyond background information and an introduction to the policy(s). The remaining three studies provided information to control groups that the intervention groups did not receive, e.g. information about the planet Pluto. Further details on the characteristics of eligible studies are reported in electronic supplementary material, tables S1 and S2.

## 3.3. Risk of bias within studies

The majority of the studies were at high risk of bias. For the evidence of effectiveness meta-analysis, 26 studies were rated as weak (high risk of bias), nine were rated as moderate, and none were rated as strong (low risk of bias). The most common reason for this was a lack of validity and reliability testing for the main outcome variable (policy support). For the evidence of ineffectiveness meta-analysis, seven studies were rated as weak and three were rated as moderate.

## 3.4. Risk of bias across studies

Examination of the evidence of effectiveness funnel plots suggested possible asymmetry (figure 2). This was confirmed by Egger's regression, $Z = 2.21$, $p = 0.027$, that suggested a risk of bias across studies. The 'trim and fill' technique was employed to account for this bias.

Examination of the evidence of ineffectiveness funnel plots did not reveal any asymmetry (figure 3). This was confirmed by Egger's regression, $Z = 1.22$, $p = 0.224$, that suggested no risk of bias across studies.

## 3.5. Main results

### 3.5.1. Communicating evidence of effectiveness

Communicating evidence that a policy was effective in achieving a target outcome increased support for the policy, SMD = 0.11, 95% CI [0.07, 0.15], $p < 0.0001$ (figure 4). Assuming a normal distribution, this is equivalent to increasing support for a policy from 50% to 54% (95% CI [53%, 56%]; the change varies with levels of baseline support as shown in table 2). There was moderate and significant heterogeneity, $Q$ (36) = 85.51, $p < 0.001$, $I^2 = 47\%$, $T = 0.071$, $T^2 = 0.005$. This suggests that the intervention effects vary more than would be expected by chance alone. Due to an asymmetrical funnel plot, the trim and fill method was used. This did not substantively change the results, SMD = 0.12, 95% CI [0.08, 0.15], $p < 0.0001$ (without trim and fill).

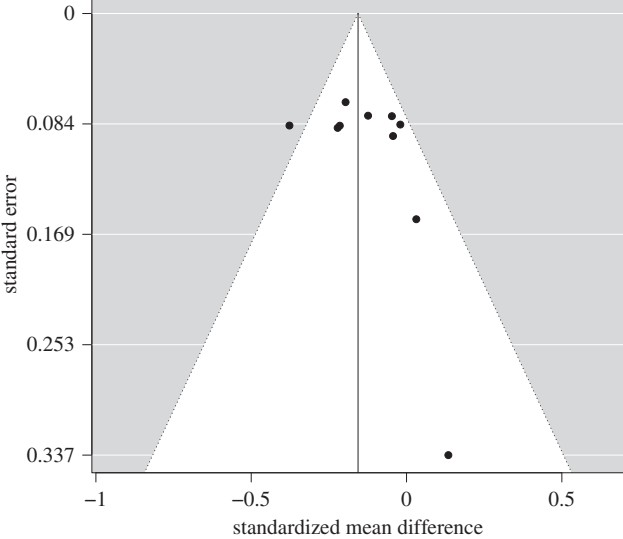

**Figure 3.** A funnel plot for the ineffectiveness meta-analysis without trim and fill.

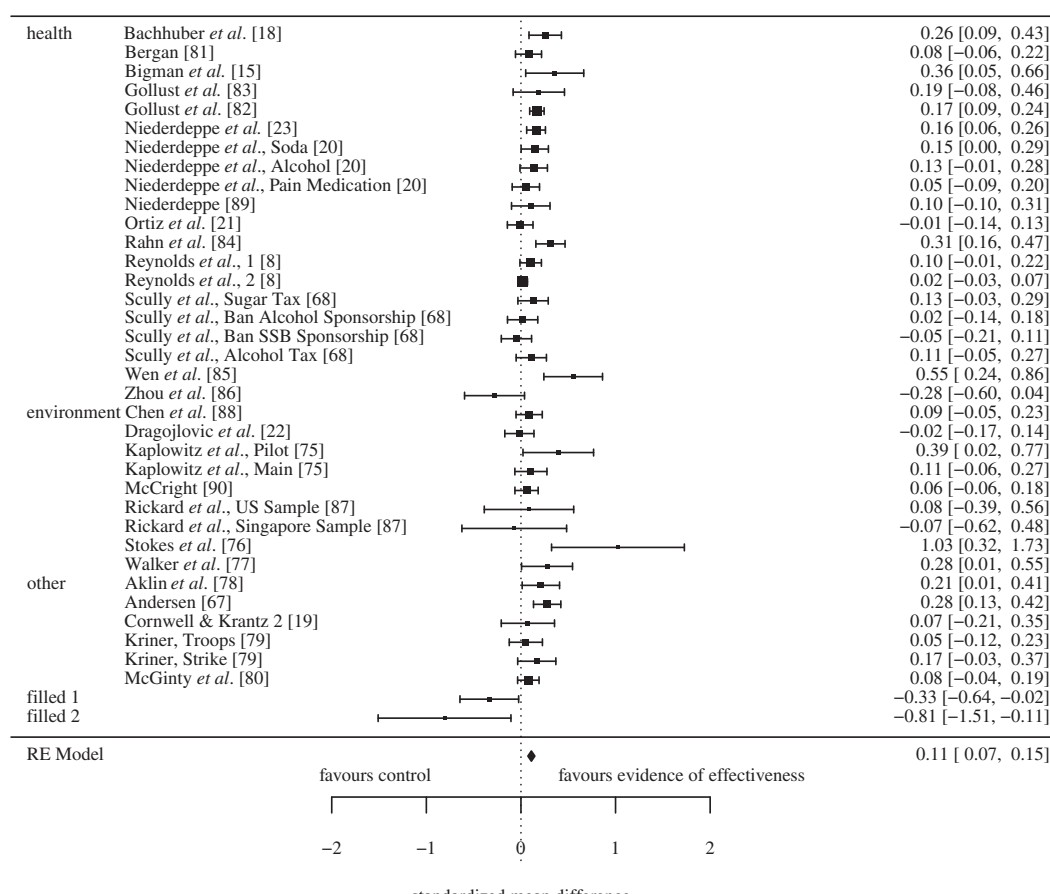

**Figure 4.** Forest plot of comparison: evidence of effectiveness communicated or no evidence of effectiveness communicated and support for the policy (corrected for bias using trim and fill).

### 3.5.1.1. Sensitivity analysis

The main analysis was re-run to test whether the significant overall effect remained after (i) excluding the studies that were at high risk of bias, and (ii) excluding studies that contained confounded interventions. Excluding studies at high risk of bias resulted in $k = 9$ effect sizes and $N = 12\,527$. Communicating evidence that a policy was effective increased support for the policy, SMD = 0.12, 95% CI [0.04, 0.20], $p = 0.002$. There was substantial and significant heterogeneity, $Q\,(8) = 27.06$, $p < 0.001$, $I^2 = 65\%$, $T = 0.09$, $T^2 = 0.01$.

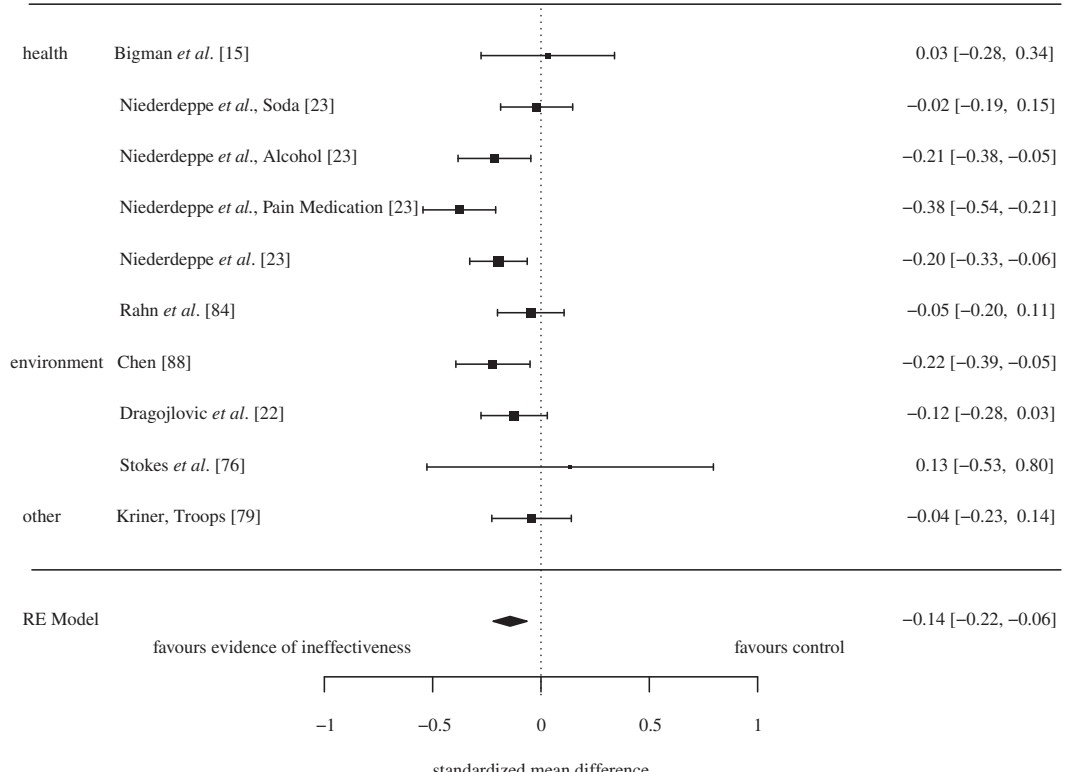

**Figure 5.** Forest plot of comparison: evidence of ineffectiveness communicated or no evidence of ineffectiveness communicated and support for the policy.

**Table 1.** Meta-regressions of the moderation analysis on the effect communicating effectiveness and policy support.

|  | B | s.e. | 95% CI | p-value |
|---|---|---|---|---|
| policy domain (reference = health)[a] | | | | |
| environment | −0.04 | 0.06 | [−0.15, 0.07] | 0.478 |
| other | 0.01 | 0.05 | [−0.09, 0.12] | 0.115 |
| presentation of effectiveness | −0.01 | 0.04 | [−0.09, 0.07] | 0.725 |
| readability | 0.01 | 0.01 | [−0.01, 0.02] | 0.311 |
| presence of uncertainty[a] | 0.04 | 0.04 | [−0.05, 0.12] | 0.394 |

[a]Study variance included in the regression due to heterogeneity.

Excluding studies that had confounded interventions resulted in $k = 11$ effect sizes and $N = 5870$. There was no funnel plot asymmetry and therefore trim and fill was not used. The effect of communicating evidence of effectiveness increased support for the policies among these studies, SMD = 0.12, 95% CI [0.04, 0.19], $p = 0.002$. There was no significant heterogeneity among these studies, $Q$ (10) = 15.58, $p = 0.112$, $I^2 = 37\%$, $T = 0.07$, $T^2 = 0.005$.

### 3.5.1.2. Moderator analyses

As seen in table 1, there was no evidence that policy domain, presentation of effectiveness, readability or the presence of uncertainty moderated the size of the effects.

### 3.5.2. Communicating evidence of ineffectiveness

Communicating evidence that a policy was ineffective or leads to undesirable outcomes decreased support for the policy, SMD = −0.14, 95% CI [−0.22, −0.06], $p < 0.001$ (figure 5). Assuming a normal distribution, this is equivalent to support for a policy decreasing from 50% to 44% (95% CI [41%,

**Table 2.** Estimated change in support (% [95% CI]) depending on (i) initial support for the policy and (ii) communication of evidence of effectiveness or ineffectiveness.

| initial support for the policy | support for the policy after communicating effectiveness | support for the policy after communicating ineffectiveness |
|---|---|---|
| 10 | 12 (11, 13) | 8 (7, 9) |
| 20 | 23 (22, 24) | 16 (14, 18) |
| 30 | 34 (33, 35) | 25 (23, 28) |
| 40 | 44 (43, 46) | 35 (32, 38) |
| 50 | 54 (53, 56) | 44 (41, 47) |
| 60 | 64 (63, 66) | 54 (51, 58) |
| 70 | 74 (73, 75) | 65 (62, 68) |
| 80 | 83 (82, 84) | 76 (73, 78) |
| 90 | 92 (91, 92) | 87 (86, 89) |

47%]; the change varies with levels of baseline support as shown in table 2). There was moderate and marginally significant heterogeneity, $Q$ (9) = 16.27, $p = 0.062$, $I^2 = 46\%$, $T = 0.08$, $T^2 = 0.007$. This suggests that the intervention effects vary marginally more so than would be expected by chance alone. There were insufficient studies to run moderation and sensitivity-by-covariate meta-analyses.

### 3.5.2.1. Sensitivity analysis

The main analysis was re-run to test whether the significant overall effect remained after excluding the studies that were at high risk of bias. Excluding studies at high risk of bias resulted in $k = 3$ effect sizes and $N = 1198$. There was no evidence that communicating evidence of ineffectiveness on policy changed support for policies, SMD = −0.08, 95% CI [−0.20, 0.03], $p = 0.155$. There was no evidence of heterogeneity, $Q$ (2) = 0.87, $p = 0.648$, $I^2 = 0\%$, $T = 0.00$, $T^2 = 0.00$.

## 4. Discussion

The results of this systematic review show that public support for a policy can be increased by communicating evidence of its effectiveness. Policies relating to health, environment and gun crime were examined, among others, and the results appear to be robust across the different domains. The results also do not significantly vary with intervention characteristics, including whether evidence of effectiveness was presented alone or confounded with other information, whether uncertainty was expressed when communicating evidence, the readability of the intervention, or whether the statement of effectiveness was asserted or described (quantitatively or qualitatively). These results also suggest that public support can be decreased by communicating evidence that the policy is ineffective or has undesirable outcomes, such as costing money to implement. Due to the small number of studies in this second meta-analysis, it was not possible to test whether the effect varies based on study and intervention characteristics. In contrast to our predictions that communicating ineffectiveness would lead to greater changes, the effect sizes estimated by these two meta-analyses were similar. With a policy that receives approximately 50% support—such as a levy on sugar-sweetened beverages [13]—communicating (in)effectiveness could increase support to 54% or decrease support to 44%.

Although the effect-size may seem relatively modest, small effects can have large real-world consequences [63]. To contextualize this, consider the role of evidence in driving public acceptability of major policy proposals, such as the UK European Union (EU) membership referendum held in 2016, where differences in public support came down to 3.8% (51.9% versus 48.1%) in favour of leaving the EU. In referenda, there is a pre-specified threshold of public support at which the government decides to implement the particular course of action. With other policy matters the threshold is not clear. While there is evidence demonstrating the importance of public support for enabling the implementation of a policy [4–6,64,65], there is uncertainty regarding the amount of support that is needed, with differences across countries to be expected. Despite this uncertainty, the current study does suggest that communicating evidence of policy effectiveness can form one part of an advocacy strategy designed to enable the implementation of effective policies. By contrast,

communicating evidence of ineffectiveness appears to be able to reduce support to a similar degree. Real-world examples of this are plentiful, with a number of media outlets around the world reporting stories that gun control does not cut gun crime, the sugar tax does not reduce obesity, or that solar power is inefficient and unreliable. These messages are likely to damage support for these policies.

The changes that were found in public support in the current study were probably driven by a change in beliefs about a policy's effectiveness. Only one study in this review examined this directly: perceived effectiveness was the mechanism behind the change in support [8]. Across three studies, Reynolds *et al.* communicated evidence that a tax on confectionery would impact childhood obesity and/or inequalities in childhood obesity. Although not all of the interventions had the desired impact on public support, the results suggest that when public support was increased, it was primarily mediated by a change in the belief that the tax was effective at reducing childhood obesity. One further study also demonstrated that communicating evidence of the policy's effectiveness increased both perceptions of policy effectiveness and support for the policy, consistent with the mechanism interpretation [15].

These results also touch on the question about whether people update their beliefs and attitudes when exposed to evidence that contradicts or supports their pre-existing beliefs. There have been concerns that when the people are given evidence, they either ignore it or become even more entrenched in their current beliefs (i.e. a backfire effect; [27–29]). Contrary to this, the current findings support recent work suggesting that people's beliefs and attitudes are somewhat amenable to change [31–33,37,38,66]. However, using the current methods, it is not possible to determine which of the participants changed their beliefs and attitudes. The results of the current study could be interpreted as showing that on average exposing people to evidence can shift their views to reflect the evidence.

## 4.1. Strengths and limitations

The current study is the first of its kind to synthesize the available evidence concerning the impact of communicating evidence of a policy's effectiveness on public support. Given the mixed results of interventions that contain evidence of effectiveness [18–20,67], the current study provides the strongest evidence yet that communicating evidence that a policy is effective can increase support for a range of different policies.

While conducting this review at least two of the authors made decisions about study inclusion, data extraction, coding and quality assessment. When the decision was ambiguous or the two authors disagreed, a third, fourth, or occasionally a fifth reviewer would enter the discussion to resolve the decision. This process minimizes the likelihood of errors that arises from single reviewer decisions [58].

The majority of the studies in the effectiveness meta-analysis (26/35) and ineffectiveness meta-analysis (7/10) were at high risk of bias. Interpretations of the pooled estimates should therefore be treated cautiously. However, despite this overall rating, there is reason to believe that this may be a harsh description of the included studies. The included studies were all randomized experiments and many included a nationally representative sample. The quality assessment tool assesses a range of different factors. In particular, one of these factors received lower quality ratings across studies: outcome validity and reliability. Only one study reported tests that their outcome measure was valid [67]. Despite the outcome measures across studies being largely similar, the validity test of this one study could not be used to validate the measures used in the other studies. Future research would benefit from improving reporting standards, including confirming the validity and reliability of policy support measures.

There was moderate heterogeneity in the primary meta-analysis and this needs to be considered when interpreting the findings. A proportion of this heterogeneity was due to interventions that include additional information (other than evidence of effectiveness) such as information about the magnitude or consequences of the problem that is being targeted. In a sensitivity analysis, interventions that only included evidence of effectiveness were analysed, and there was no significant heterogeneity. This suggests that the effect sizes are consistent when communicating messages solely about policy effectiveness.

There was some evidence of funnel plot asymmetry for the evidence of effectiveness meta-analysis but not for the evidence of ineffectiveness meta-analysis. This may indicate publication bias, but other sources of bias could also explain this, such as language bias and outcome reporting bias [58]. We addressed this by using the 'trim and fill' approach. While this does not eradicate the problem it can reduce it.

The majority of the primary research included in this review was conducted in the USA (25/36) and the majority of policies studied were in the domain of health (20/36). The results of the review may therefore be less generalizable to communicating evidence of effectiveness for policies other than health in countries other than the USA. However, in the meta-regression we did not find evidence

that policy domain moderated the effect of communicating evidence but we note that this analysis had too low power to fully exclude this possibility.

## 4.2. Future research

Current studies preclude an estimate of effects in real-world settings, where multiple and competing messages are likely to be present. If a government desired to implement policies then competing messages may originate from a variety of sources. The results of the ineffectiveness meta-analysis suggest that opponents of policy can find success in communicating evidence that a policy is ineffective, which leads to reductions in support that are comparable in magnitude to communicating evidence of effectiveness. Online experiments have attempted to simulate how people respond to competing messages [20,68,69], and while this is a step forward in examining how competing messages are interpreted, this does not replicate how information is processed in the real world. Building on the work of Niederdeppe *et al*. would be a useful direction for this goal.

The meta-regressions conducted here suggested that the effect of communicating evidence on public support was not moderated by several intervention characteristics. However, this was based on associations rather than experimentation, and with a relatively small number of included studies, and a large number of confounding variables. Further research should investigate optimal communication methods from the perspectives of comprehension, accuracy and successful belief change [8,48]. One aspect of optimal communication may involve audience segmentation or targeted communication [70]. Personalized risk communication for changing behaviour has been shown to be either ineffective or of little effectiveness [71,72]. Nonetheless, psychographic targeting may have potential [73]. This approach suggests that tailoring a communication to the recipient's personality may increase the magnitude of any change. Currently this approach lacks a solid evidence base. Additionally it has recently come under ethical scrutiny [74]. More acceptable approaches would be those that enable recipients to make more informed decisions rather than to simply persuade them one way or another.

## 5. Conclusion

In summary, this review provides the most robust evidence to date that communicating evidence that a policy is effective can increase support for the policy, and comparable decreases in support can be achieved by communicating evidence that the policy is not effective. These changes in support may typically be considered as small, yet could be meaningful when considered at the population level. Presidential elections and referenda decisions have hung on less.

Data accessibility. The data and code are available at https://osf.io/4gjur/.

Authors' contributions. J.P.R. participated in the design of the study, participated in the screening of studies, participated in data analysis, interpreted the results and drafted the manuscript. K.S. participated in the design of the study, searched the databases, participated in the screening of studies, interpreted the results and gave comments on the manuscript. M.P. participated in data analysis and gave comments on the manuscript. S.v.d.L. participated in the design of the study, interpreted the results and gave comments on the manuscript. T.M.M. conceived the initial idea, participated in the design of the study, interpreted the results and gave comments on the manuscript. All authors gave final approval for publication.

Competing interests. The authors do not have any competing interests.

Funding. This report is independent research commissioned and funded by the National Institute for Health Research Policy Research Programme (Policy Research Unit in Behaviour and Health (PR-UN-0409-10109)). The views expressed in this publication are those of the authors and not necessarily those of the NHS, the National Institute for Health Research, the Department of Health and Social Care or its arm's length bodies, and other Government Departments.

Acknowledgements. The authors are grateful to Eleni Mantzari and Milica Vasiljevic for critical comments on an earlier draft of the paper.

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
