## [Reviewer comments · Royal Society Open Science]

Review History

RSOS-190522.R0 (Original submission)

Review form: Reviewer 1

Is the manuscript scientifically sound in its present form?

Yes

Are the interpretations and conclusions justified by the results?

Yes

Is the language acceptable?

Yes

Is it clear how to access all supporting data?

Yes

Do you have any ethical concerns with this paper?

No

Have you any concerns about statistical analyses in this paper?

Yes

Recommendation?

Accept with minor revision (please list in comments)

Comments to the Author(s)

Referee report for: "Communicating the effectiveness and ineffectiveness of government policies and its impact on public support: A systematic review and meta-analysis" (RSOS-190522)

This meta-analysis confirms a commonsense, good-news result. Providing information about policy effectiveness increases support for a policy and providing information about ineffectiveness decreases support. The effect in each direction are about the same magnitude.

I support publication of this article, subject to some revisions.

1) Please remove the discussion of study quality. In my view, neither of the two reasons given for marking a study as being poor quality are appropriate.

First, "not reporting whether groups were matched for confounding variables" was given as a reason to think an average treatment effect estimate was prone to bias. "Matching for confounding variables" is presumably about blocking. The purpose of blocking is to increase precision, not decrease bias (Gerber and Green, chapter 3). Because the experiments were randomized, bias is by design zero, even if the experiment is not blocked (obviously not every estimate is exactly equal to the estimand. But bias is a property of a design that, on average, over all possible random assignments, the average of the estimates is equal to the estimand). Blocking has the advantage of decreasing standard errors, but it does not improve bias... precisely because experiments that use, say, Bernoulli or complete random assignment generate unbiased estimates too. Please do not rate studies as weak because they did not block.

Second, "a lack of of validity and reliability testing for the main outcome variable (policy support)" was given as a reason rate a study as weak. In my view, this is inappropriate. While I agree that the quality of outcome measures varies from study to study, I do not agree that a coding of whether the authors engaged in validity or reliability testing is itself a good measure of outcome quality. To begin with, existing procedures for demonstrating validity are, in my view, quite weak, and I would only in rare occasions conclude on the basis of such procedures that my outcome measure was high quality. Demonstrations that the sort of policy support variables considered here are reliable (i.e., are low variance) are mostly irrelevant because the estimand is the average treatment effect of policy information. Extra noise in the outcome variable just increases standard errors, but that weakness is appropriately reflected in the standard error of the estimate. In my view, arguments in favor of an outcome variable should have to be made on qualitative, theoretical grounds. Please do not rate the quality of studies according to the arbitrary measure of whether they engaged in reliability and validity testing (whose validity I wholeheartedly question!).

In my view, the biggest quality distinction is between randomized and nonrandomized studies, because of the threat of selection into treatment in nonrandomized studies. But because none of those studies made it into this meta-analysis, the set of studies included all provide reasonable estimates of the average effect of policy (in)effectiveness on support. I would therefore contend these studies are mostly of comparable quality. On this basis, I would ask that the authors to cut the discussion of study quality altogether.

2) The choice of random-effects meta analysis makes sense to me because we have no ex-ante reason to think that this very heterogeneous set of studies, all of which use different outcome measures and different treatments would be estimating the *EXACT SAME PARAMETER*, as would be assumed by a fixed-effects approach. What's odd to me is the other justifications given for random effects, e.g.: "This suggests that the intervention effects vary more so than would be expected by chance alone and therefore justifies using a random-effects model." The *results* do not justify the choice of model -- the theoretical setting does!

I don't understand why the model switches from maximum likelihood to Bayesian only in cases where the results indicate strong heterogeneity across studies. The Bayesian model works just as well when the studies aren't heterogeneous. Please pick a statistical model, state it, and stick with it.

The random effects model explicitly assumes that the true ATEs estimated in each study are drawn from a normal distribution with mean μ and standard deviation τ . Right now, the meta analysis focuses on μ and associated inferential statistics. The discussion of heterogeneity describes a set of statistics that are mostly focused on whether the amount of heterogeneity is statistically significant. I'm pretty much uninterested in whether the heterogeneity is statistically significant, though I can imagine that some people are interested in it. For me, however, I think the τ parameter itself -- exactly the measure of how dispersed effects are -- describes heterogeneity best. Please report estimates of τ and interpret them for the reader.

Smaller comments

1. I was unfamiliar with Egger's regression and trim-and-fill, so I had to look them up. Thank you for the chance to learn about them! I think a sentence or two describing the purpose of these procedures and what problem they try to solve would help readers who like me were unfamiliar with them.

2. Style comments: (a) "There are many problems currently facing nations around the world." is a weak opening sentence. (b) I doubt this primacy claim: "The current study aims to provide the first systematic synthesis of the evidence for changing public attitudes and support for policies by communicating evidence about a policy's effectiveness at achieving its goal." (c) lots of extra parentheticals and i.e.,'s and e.g.,'s that could be smoothed out.

3. When a study has two outcome variables and the mean effect is used, how do you calculate the standard error / confidence interval? The two estimates certainly aren't independent, and we can't just take the mean of the standard errors/ confidence intervals. Were these calculated from the raw data and not from summary statistics?

4. "the intervention that offered the least confounded version of a treatment was selected (e.g., an intervention containing evidence of effectiveness would be chosen over an intervention containing evidence of effectiveness and information on the nature of the problem)" Because the word confounded has a specific statistical meaning that is not what is meant here, I would suggest the authors reword.

5. This sentence inappropriately slips into causal language: "none of the moderators significantly affected the size of the effects." Predicts? Associated? [I'd also mention the low power of such tests for plausibly-sized differences in effect size]

6. I like that I got to look at the code and data. Thank you!

Review form: Reviewer 2

Is the manuscript scientifically sound in its present form?

Yes

Are the interpretations and conclusions justified by the results?

Yes

Is the language acceptable?

Yes

Is it clear how to access all supporting data?

Yes

Do you have any ethical concerns with this paper?

No

Have you any concerns about statistical analyses in this paper?

I do not feel qualified to assess the statistics

Recommendation?

Accept with minor revision (please list in comments)

Comments to the Author(s)

I think this is a neatly written paper, and as it is a first systematic contribution to a relevant research question, it has merits.

I have, however, some concerns which I could not dismiss by reading the paper. These might be due to my insufficient knowledge of meta-analytic methods, but it would be appreciated if the authors could address them in a revised version or, if they are not relevant, by simply saying so in their reply and explaining why.

1) It is somehow surprising that all eligible studies are randomized experiments. If this is true, it is an interesting result by itself... No one has ever explored ex-post, with real data, how an information campaign on the effectiveness of a policy has influenced public support? How confident are the authors that only hypothetical survey with experimental information provision exist?

2) The assumption is that participants rely on the information on effectiveness provided within the study, but they may be aware of other/conflictual information. How this is controlled for in the various studies?

3) While not being an expert, I am aware that meta-analysis is 'acceptable' with as few as two studies. However, I struggle with this concept, especially if one wants to control for study heterogeneity (e.g. through random effects). While there is obviously value in the exercise, I wonder to what extent a limited number of studies (35) is credible to generalize the findings of this research. Even if this might be trivial for meta-analysis specialists, a more explicit discussion on why this small number of (all experimental) studies produces generalisable findings would be appreciated, together with a discussion of the limitations. Furthermore, these 35 studies are from 23 different authors if I have computed correctly (one author appears five times, another three times, and two papers are from the lead author of the current study).

4) Why aren't funnel plots displayed in the paper? A graph of estimates and precision of the included studies would be very informative.

5) Not unrelated to the points (3) and (4) is the risk of publication bias. This is mentioned by the authors, and I think that section is valuable... but the discussion is too short and "implicit" to be

understood. And how can one elicit publication bias from the funnel plots? How many journals publish negative or null findings from experiments?

6) "Conversion from the pooled effect sizes to changes in policy support proportions were calculated using a number needed to treat formula (see Table 2; [61])". What does this mean? Can the author be more explicit about their data processing?

7) I wonder how standard errors are treated in this meta analysis. The authors report large Ns by considering the sample size of the analysed studies, but the number of studies is small. I am not sure that allowing for random effects does the full job. Is there some form of clustering for the standard errors? I would also be curious to see the results from a fixed effect model as a robustness check, I don't see particular reasons why the results should be different?

8) Table 2 is extremely relevant and impressive in terms of its 'linearity' and precision... but it's not clear to me how it was estimated, given the extremely low number of studies. I am especially surprised by the relatively narrow confidence intervals. Was a linear effect assumed? See also my comment on clustering above.

9) (Minor) Page 19 - confectionery instead of confectionary?

Decision letter (RSOS-190522.R0)

30-Jul-2019

Dear Dr Reynolds

On behalf of the Editors, I am pleased to inform you that your Manuscript RSOS-190522 entitled "Communicating the effectiveness and ineffectiveness of government policies and its impact on public support: A systematic review and meta-analysis" has been accepted for publication in Royal Society Open Science subject to minor revision in accordance with the referee suggestions. Please find the referees' comments at the end of this email.

The reviewers and handling editors have recommended publication, but also suggest some minor revisions to your manuscript. Therefore, I invite you to respond to the comments and revise your manuscript.

- Ethics statement

- Data accessibility

If you wish to submit your supporting data or code to Dryad (<http://datadryad.org/>), or modify your current submission to dryad, please use the following link:
<http://datadryad.org/submit?journalID=RSOS&manu=RSOS-190522>

- **Competing interests**

- **Authors' contributions**

- **Acknowledgements**

- **Funding statement**

Because the schedule for publication is very tight, it is a condition of publication that you submit the revised version of your manuscript before 08-Aug-2019. Please note that the revision deadline will expire at 00.00am on this date. If you do not think you will be able to meet this date please let me know immediately.

When submitting your revised manuscript, you will be able to respond to the comments made by the referees and upload a file "Response to Referees" in "Section 6 - File Upload". You can use this to document any changes you make to the original manuscript. In order to expedite the

processing of the revised manuscript, please be as specific as possible in your response to the referees. We strongly recommend uploading two versions of your revised manuscript:

Kind regards,
Alice Power
Editorial Coordinator
Royal Society Open Science

on behalf of Dr Christina Demski (Associate Editor) and Essi Viding (Subject Editor)
openscience@royalsociety.org

Associate Editor Comments to Author (Dr Christina Demski):

Both reviewers agree that the manuscript deserves publication, but request revisions before it can be published. Indeed, both reviewers have quite a few questions about the methodological and statistical choices made. Please respond fully to each of them. I look forward to your response.

Reviewer comments to Author:

Reviewer: 1

Comments to the Author(s)

Referee report for: "Communicating the effectiveness and ineffectiveness of government policies and its impact on public support: A systematic review and meta-analysis" (RSOS-190522)

This meta-analysis confirms a commonsense, good-news result. Providing information about policy effectiveness increases support for a policy and providing information about ineffectiveness decreases support. The effect in each direction are about the same magnitude.

I support publication of this article, subject to some revisions.

1) Please remove the discussion of study quality. In my view, neither of the two reasons given for marking a study as being poor quality are appropriate.

First, "not reporting whether groups were matched for confounding variables" was given as a reason to think an average treatment effect estimate was prone to bias. "Matching for confounding variables" is presumably about blocking. The purpose of blocking is to increase precision, not decrease bias (Gerber and Green, chapter 3). Because the experiments were randomized, bias is by design zero, even if the experiment is not blocked (obviously not every estimate is exactly equal to the estimand. But bias is a property of a design that, on average, over all possible random assignments, the average of the estimates is equal to the estimand). Blocking has the advantage of decreasing standard errors, but it does not improve bias... precisely because experiments that use, say, Bernoulli or complete random assignment generate unbiased estimates too. Please do not rate studies as weak because they did not block.

Second, "a lack of validity and reliability testing for the main outcome variable (policy support)" was given as a reason rate a study as weak. In my view, this is inappropriate. While I agree that the quality of outcome measures varies from study to study, I do not agree that a coding of whether the authors engaged in validity or reliability testing is itself a good measure of outcome quality. To begin with, existing procedures for demonstrating validity are, in my view, quite weak, and I would only in rare occasions conclude on the basis of such procedures that my outcome measure was high quality. Demonstrations that the sort of policy support variables considered here are reliable (i.e., are low variance) are mostly irrelevant because the estimand is the average treatment effect of policy information. Extra noise in the outcome variable just increases standard errors, but that weakness is appropriately reflected in the standard error of the estimate. In my view, arguments in favor of an outcome variable should have to be made on qualitative, theoretical grounds. Please do not rate the quality of studies according to the

arbitrary measure of whether they engaged in reliability and validity testing (whose validity I wholeheartedly question!).

In my view, the biggest quality distinction is between randomized and nonrandomized studies, because of the threat of selection into treatment in nonrandomized studies. But because none of those studies made it into this meta-analysis, the set of studies included all provide reasonable estimates of the average effect of policy (in)effectiveness on support. I would therefore contend these studies are mostly of comparable quality. On this basis, I would ask that the authors to cut the discussion of study quality altogether.

2) The choice of random-effects meta analysis makes sense to me because we have no ex-ante reason to think that this very heterogeneous set of studies, all of which use different outcome measures and different treatments would be estimating the *EXACT SAME PARAMETER*, as would be assumed by a fixed-effects approach. What's odd to me is the other justifications given for random effects, e.g.: "This suggests that the intervention effects vary more so than would be expected by chance alone and therefore justifies using a random-effects model." The *results* do not justify the choice of model -- the theoretical setting does!

I don't understand why the model switches from maximum likelihood to Bayesian only in cases where the results indicate strong heterogeneity across studies. The Bayesian model works just as well when the studies aren't heterogeneous. Please pick a statistical model, state it, and stick with it.

The random effects model explicitly assumes that the true ATEs estimated in each study are drawn from a normal distribution with mean μ and standard deviation τ . Right now, the meta analysis focuses on μ and associated inferential statistics. The discussion of heterogeneity describes a set of statistics that are mostly focused on whether the amount of heterogeneity is statistically significant. I'm pretty much uninterested in whether the heterogeneity is statistically significant, though I can imagine that some people are interested in it. For me, however, I think the τ parameter itself -- exactly the measure of how dispersed effects are -- describes heterogeneity best. Please report estimates of τ and interpret them for the reader.

Smaller comments

1. I was unfamiliar with Egger's regression and trim-and-fill, so I had to look them up. Thank you for the chance to learn about them! I think a sentence or two describing the purpose of these procedures and what problem they try to solve would help readers who like me were unfamiliar with them.

2. Style comments: (a) "There are many problems currently facing nations around the world." is a weak opening sentence. (b) I doubt this primacy claim: "The current study aims to provide the first systematic synthesis of the evidence for changing public attitudes and support for policies by communicating evidence about a policy's effectiveness at achieving its goal." (c) lots of extra parentheticals and i.e.,'s and e.g.,'s that could be smoothed out.

3. When a study has two outcome variables and the mean effect is used, how do you calculate the standard error / confidence interval? The two estimates certainly aren't independent, and we can't just take the mean of the standard errors/ confidence intervals. Were these calculated from the raw data and not from summary statistics?

4. "the intervention that offered the least confounded version of a treatment was selected (e.g., an intervention containing evidence of effectiveness would be chosen over an intervention containing evidence of effectiveness and information on the nature of the problem)" Because the

word confounded has a specific statistical meaning that is not what is meant here, I would suggest the authors reword.

5. This sentence inappropriately slips into causal language: "none of the moderators significantly affected the size of the effects." Predicts? Associated? [I'd also mention the low power of such tests for plausibly-sized differences in effect size]

6. I like that I got to look at the code and data. Thank you!

Reviewer: 2

Comments to the Author(s)

I think this is a neatly written paper, and as it is a first systematic contribution to a relevant research question, it has merits.

I have, however, some concerns which I could not dismiss by reading the paper. These might be due to my insufficient knowledge of meta-analytic methods, but it would be appreciated if the authors could address them in a revised version or, if they are not relevant, by simply saying so in their reply and explaining why.

1) It is somehow surprising that all eligible studies are randomized experiments. If this is true, it is an interesting result by itself... No one has ever explored ex-post, with real data, how an information campaign on the effectiveness of a policy has influenced public support? How confident are the authors that only hypothetical survey with experimental information provision exist?

2) The assumption is that participants rely on the information on effectiveness provided within the study, but they may be aware of other/conflictual information. How this is controlled for in the various studies?

3) While not being an expert, I am aware that meta-analysis is 'acceptable' with as few as two studies. However, I struggle with this concept, especially if one wants to control for study heterogeneity (e.g. through random effects). While there is obviously value in the exercise, I wonder to what extent a limited number of studies (35) is credible to generalize the findings of this research. Even if this might be trivial for meta-analysis specialists, a more explicit discussion on why this small number of (all experimental) studies produces generalisable findings would be appreciated, together with a discussion of the limitations. Furthermore, these 35 studies are from 23 different authors if I have computed correctly (one author appears five times, another three times, and two papers are from the lead author of the current study).

4) Why aren't funnel plots displayed in the paper? A graph of estimates and precision of the included studies would be very informative.

5) Not unrelated to the points (3) and (4) is the risk of publication bias. This is mentioned by the authors, and I think that section is valuable... but the discussion is too short and "implicit" to be understood. And how can one elicit publication bias from the funnel plots? How many journals publish negative or null findings from experiments?

6) "Conversion from the pooled effect sizes to changes in policy support proportions were calculated using a number needed to treat formula (see Table 2; [61])". What does this mean? Can the author be more explicit about their data processing?

7) I wonder how standard errors are treated in this meta analysis. The authors report large Ns by considering the sample size of the analysed studies, but the number of studies is small. I am not sure that allowing for random effects does the full job. Is there some form of clustering for the standard errors? I would also be curious to see the results from a fixed effect model as a robustness check, I don't see particular reasons why the results should be different?

8) Table 2 is extremely relevant and impressive in terms of its 'linearity' and precision... but it's not clear to me how it was estimated, given the extremely low number of studies. I am especially surprised by the relatively narrow confidence intervals. Was a linear effect assumed? See also my comment on clustering above.

9) (Minor) Page 19 - confectionery instead of confectionary?

Author's Response to Decision Letter for (RSOS-190522.R0)

See Appendix A.

RSOS-190522.R1 (Revision)

Review form: Reviewer 1

Is the manuscript scientifically sound in its present form?

Yes

Are the interpretations and conclusions justified by the results?

Yes

Is the language acceptable?

No

Do you have any ethical concerns with this paper?

No

Have you any concerns about statistical analyses in this paper?

Yes

Recommendation?

Accept with minor revision (please list in comments)

Comments to the Author(s)

I am grateful to the authors to for their response to my point (2) about the choice of statistical model. I also agree with the changes they made in response to my smaller comments. The paper is now in my view very nearly ready for publication, though I must say I am not convinced by the authors' rebuttal to my critique (1).

The main sticking point appears to be about bias due to chance imbalances that crop up in some randomizations. The authors describe studies as "weak" if they do not discuss covariate adjustment, or matching, or other adjustment for pre-treatment covariates. In their response memo, the authors justify this choice by pointing to references that describe how when covariates are out of balance, estimates tend to be further from the truth. That claim is true with respect to a particular estimate, but it is NOT true with respect to an estimator. The estimators are unbiased regardless of covariate adjustment, though in the absence of adjustment, estimates can be far from the estimand.

Indeed, the *main point* of a randomized experiment is to provide an unbiased procedure that gets the answer right on average. Bias is defined as a property of the procedure over the set of possible randomizations, NOT as a property of the distance between the estimate and the estimand in any single realization. When we do covariate adjustment, it's not to correct bias, it's to reduce variance, i.e., by systematically drawing estimates that are far from the truth in towards the truth. This is not about bias -- both the unadjusted and adjusted procedures are (approximately) unbiased for the ATE. This paper should emphatically not claim that "not reporting whether groups were matched for confounding variables was also a large factor". In a randomized trial, this consideration is about precision, not bias. I care about this point, and I would encourage the authors to please reconsider their position.

I also do not think that it is appropriate to grade studies as weak because they didn't use a measurement scale that had been specifically validated. Those that have may still be quite weak, and those that haven't may be quite strong. This quality discussion, I think, is misplaced. I think the paper would be improved by removing the "Risk of Bias within studies" altogether.

One final point. The authors write in their rebuttal that the variance estimates for pooled are more conservative than if the authors had used estimates that included the correlation across studies. Is this a worst-case variance estimator? If so, I think the text needs more detail on that procedure. Most naive approaches I can think of would not necessarily be conservative. In any case, I can't tell from the text what that variance estimator is, so I can't evaluate the claim that it is conservative. Some clarification on this would be appreciated.

Those remaining sticking points notwithstanding, I think this is a very nice meta analysis that I'll be citing in the future. My thanks to the authors.

Review form: Reviewer 2

Is the manuscript scientifically sound in its present form?

Yes

Are the interpretations and conclusions justified by the results?

Yes

Is the language acceptable?

Yes

Do you have any ethical concerns with this paper?

No

Have you any concerns about statistical analyses in this paper?

No

Recommendation?

Accept as is

Comments to the Author(s)

Thanks for addressing my comments in a neat and accurate way. I have no further concerns, this seems a relevant contribution to the literature.

Decision letter (RSOS-190522.R1)

04-Nov-2019

Dear Dr Reynolds:

On behalf of the Editors, I am pleased to inform you that your Manuscript RSOS-190522.R1 entitled "Communicating the effectiveness and ineffectiveness of government policies and its impact on public support: A systematic review and meta-analysis" has been accepted for publication in Royal Society Open Science subject to minor revision in accordance with the referee suggestions. Please find the referees' comments at the end of this email.

The reviewers and Subject Editor have recommended publication, but also suggest some minor revisions to your manuscript. Therefore, I invite you to respond to the comments and revise your manuscript.

- Ethics statement

- Data accessibility

If you wish to submit your supporting data or code to Dryad (<http://datadryad.org/>), or modify your current submission to dryad, please use the following link:
<http://datadryad.org/submit?journalID=RSOS&manu=RSOS-190522.R1>

- Competing interests

- Authors' contributions

- Acknowledgements

- Funding statement

Because the schedule for publication is very tight, it is a condition of publication that you submit the revised version of your manuscript before 13-Nov-2019. Please note that the revision deadline will expire at 00.00am on this date. If you do not think you will be able to meet this date please let me know immediately.

Supplementary files will be published alongside the paper on the journal website and posted on

the online figshare repository (<https://figshare.com>). The heading and legend provided for each supplementary file during the submission process will be used to create the figshare page, so please ensure these are accurate and informative so that your files can be found in searches. Files on figshare will be made available approximately one week before the accompanying article so that the supplementary material can be attributed a unique DOI.

on behalf of Dr Christina Demski (Associate Editor) and Essi Viding (Subject Editor)
openscience@royalsociety.org

Associate Editor Comments to Author (Dr Christina Demski):

The reviewers have now looked at your revised manuscript and find it almost ready for publication. Reviewer 2 provides a favorable review of the manuscript but would like a couple of more clarifications. In addition, the reviewer follows up with some additional thoughts on one of his main points from the first review, which we would like you to respond to.

Reviewer comments to Author:
Reviewer: 2

Comments to the Author(s)

Thanks for addressing my comments in a neat and accurate way. I have no further concerns, this seems a relevant contribution to the literature.

Reviewer: 1

Comments to the Author(s)

I am grateful to the authors to for their response to my point (2) about the choice of statistical model. I also agree with the changes they made in response to my smaller comments. The paper is now in my view very nearly ready for publication, though I must say I am not convinced by the authors' rebuttal to my critique (1).

The main sticking point appears to be about bias due to chance imbalances that crop up in some randomizations. The authors describe studies as "weak" if they do not discuss covariate adjustment, or matching, or other adjustment for pre-treatment covariates. In their response memo, the authors justify this choice by pointing to references that describe how when covariates are out of balance, estimates tend to be further from the truth. That claim is true with respect to a particular estimate, but it is NOT true with respect to an estimator. The estimators are unbiased regardless of covariate adjustment, though in the absence of adjustment, estimates can be far from the estimand.

Indeed, the *main point* of a randomized experiment is to provide an unbiased procedure that gets the answer right on average. Bias is defined as a property of the procedure over the set of possible randomizations, NOT as a property of the distance between the estimate and the estimand in any single realization. When we do covariate adjustment, it's not to correct bias, it's to reduce variance, i.e., by systematically drawing estimates that are far from the truth in towards the truth. This is not about bias -- both the unadjusted and adjusted procedures are (approximately) unbiased for the ATE. This paper should emphatically not claim that "not reporting whether groups were matched for confounding variables was also a large factor". In a randomized trial, this consideration is about precision, not bias. I care about this point, and I would encourage the authors to please reconsider their position.

I also do not think that it is appropriate to grade studies as weak because they didn't use a measurement scale that had been specifically validated. Those that have may still be quite weak, and those that haven't may be quite strong. This quality discussion, I think, is misplaced. I think the paper would be improved by removing the "Risk of Bias within studies" altogether.

One final point. The authors write in their rebuttal that the variance estimates for pooled are more conservative than if the authors had used estimates that included the correlation across studies. Is this a worst-case variance estimator? If so, I think the text needs more detail on that procedure. Most naive approaches I can think of would not necessarily be conservative. In any case, I can't tell from the text what that variance estimator is, so I can't evaluate the claim that it is conservative. Some clarification on this would be appreciated.

Those remaining sticking points notwithstanding, I think this is a very nice meta analysis that I'll be citing in the future. My thanks to the authors.

Author's Response to Decision Letter for (RSOS-190522.R1)

See Appendix B.

Decision letter (RSOS-190522.R2)

13-Nov-2019

Dear Dr Reynolds,

It is a pleasure to accept your manuscript entitled "Communicating the effectiveness and ineffectiveness of government policies and its impact on public support: A systematic review with meta-analysis" in its current form for publication in Royal Society Open Science. The comments of the reviewer(s) who reviewed your manuscript are included at the foot of this letter.

Please ensure that you send to the editorial office an editable version of your accepted manuscript, and individual files for each figure and table included in your manuscript. You can send these in a zip folder if more convenient. Failure to provide these files may delay the

processing of your proof. You may disregard this request if you have already provided these files to the editorial office.

on behalf of Dr Christina Demski (Associate Editor) and Essi Viding (Subject Editor)
openscience@royalsociety.org

Appendix A

We thank the editor and both reviewers for their careful reading and helpful comments on our paper. We describe our responses to these comments in detail below and very much hope that we have satisfactorily addressed them and that the manuscript is now ready to be accepted for publication in Royal Society Open Science.

Reviewer comments to Author:

Reviewer: 1

Comments to the Author(s)

Referee report for: "Communicating the effectiveness and ineffectiveness of government policies and its impact on public support: A systematic review and meta-analysis" (RSOS-190522)

This meta-analysis confirms a commonsense, good-news result. Providing information about policy effectiveness increases support for a policy and providing information about ineffectiveness decreases support. The effect in each direction are about the same magnitude.

I support publication of this article, subject to some revisions.

1) Please remove the discussion of study quality. In my view, neither of the two reasons given for marking a study as being poor quality are appropriate.

First, "not reporting whether groups were matched for confounding variables" was given as a reason to think an average treatment effect estimate was prone to bias. "Matching for confounding variables" is presumably about blocking. The purpose of blocking is to increase precision, not decrease bias (Gerber and Green, chapter 3). Because the experiments were randomized, bias is by design zero, even if the experiment is not blocked (obviously not every estimate is exactly equal to the estimate. But bias is a property of a design that, on average, over all possible random assignments, the average of the estimates is equal to the estimate). Blocking has the advantage of decreasing standard errors, but it does not improve bias... precisely because experiments that use, say, Bernoulli or complete random assignment generate unbiased estimates too. Please do not rate studies as weak because they did not block.

Second, "a lack of validity and reliability testing for the main outcome variable (policy support)" was given as a reason rate a study as weak. In my view, this is inappropriate. While I agree that the quality of outcome measures varies from study to study, I do not agree that a coding of whether the authors engaged in validity or reliability testing is itself a good measure of outcome quality. To begin with, existing procedures for demonstrating validity are, in my view, quite weak, and I would only in rare occasions conclude on the basis of such procedures that my outcome measure was high quality. Demonstrations that the sort of policy support variables considered

here are reliable (i.e., are low variance) are mostly irrelevant because the estimand is the average treatment effect of policy information. Extra noise in the outcome variable just increases standard errors, but that weakness is appropriately reflected in the standard error of the estimate. In my view, arguments in favor of an outcome variable should have to be made on qualitative, theoretical grounds. Please do not rate the quality of studies according to the arbitrary measure of whether they engaged in reliability and validity testing (whose validity I wholeheartedly question!).

In my view, the biggest quality distinction is between randomized and nonrandomized studies, because of the threat of selection into treatment in nonrandomized studies. But because none of those studies made it into this meta-analysis, the set of studies included all provide reasonable estimates of the average effect of policy (in)effectiveness on support. I would therefore contend these studies are mostly of comparable quality. On this basis, I would ask that the authors to cut the discussion of study quality altogether.

Overall the comments from the reviewer are well received. Importantly, the quality assessment was used *only* in a sensitivity analysis. The primary analyses use all studies *regardless* of their adjudged quality.

We believe that this should mitigate any substantive concerns about the treatment of quality and risk of bias in our analyses.

Moreover, we disagree that ratings of quality or risk of bias should be removed entirely. Such an assessment of risk of bias is a fundamental part of systematic review methods. It is recommended by numerous guides and is part of the gold standard Cochrane methods for conducting and reporting systematic reviews (Higgins & Green, 2008; *Cochrane handbook for systematic reviews of interventions*).

Below we respond to some of the specific points this reviewer makes.

- (a) Study quality is assessed using a multi-faceted measure of which balanced confounders and outcome validity/reliability are *just* two categories. We have now added discussion to the paper to clarify this point more explicitly (p.19):

“The quality assessment tool assesses a range of different factors. In particular, two of these received lower quality ratings across studies namely; a) matching for confounding variables and b) outcome validity and reliability”

- (b) The classification of some studies as being at risk of bias due to unbalanced covariates was not because they did not block, but rather, because they did not report that known confounders were matched. An unbalanced allocation can occur even when randomisation is used. The issue we address is whether authors have stated whether there are known covariates which could introduce bias if not allocated equally (e.g. using blocking or minimisation), and what they say they have done to address this.

- (c) Random allocation does not guarantee to balanced confounders all the time (see Deaton & Cartwright, 2018; <https://doi.org/10.1016/j.socscimed.2017.12.005>). In any trial, it is possible that by chance alone, the gender ratio (for example) may be imbalanced across treatment arms particularly in smaller samples. Routine checking and reporting of relevant

confounding variables is recommended (e.g., Moher et al., 2010; <https://doi.org/10.1016/j.jclinepi.2010.03.004>) and is therefore part of the assessment tool.

- (d) The outcome assessment is not solely based on whether study authors conducted validity or reliability testing; other sources were considered too. For example, if the study authors used measures that have been independently tested for reliability and validity by another study, then this would be sufficient. However, we looked, and none of the individual papers cited such a paper, nor did we find one by manually searching for one.
- (e) We agree that – all other things being equal - randomised studies are of higher quality than nonrandomised ones. Indeed, this distinction is included in the quality assessment tool that we use. As no non-randomised studies were included in the current review, all studies were rated as “high quality” for research design subsection of the assessment tool.

2) The choice of random-effects meta analysis makes sense to me because we have no ex-ante reason to think that this very heterogeneous set of studies, all of which use different outcome measures and different treatments would be estimating the *EXACT SAME PARAMETER*, as would be assumed by a fixed-effects approach. What's odd to me is the other justifications given for random effects, e.g.: "This suggests that the intervention effects vary more so than would be expected by chance alone and therefore justifies using a random-effects model." The *results* do not justify the choice of model -- the theoretical setting does!

Thank you for this point, we have removed this line accordingly.

I don't understand why the model switches from maximum likelihood to Bayesian only in cases where the results indicate strong heterogeneity across studies. The Bayesian model works just as well when the studies aren't heterogeneous. Please pick a statistical model, state it, and stick with it.

Apologies for any confusion, but Bayesian analyses were planned for both main outcomes (as sensitivity analyses) irrespective of the data. We have moved these Bayesian analyses to the supplementary material in order to improve the flow of the paper.

The random effects model explicitly assumes that the true ATEs estimated in each study are drawn from a normal distribution with mean μ and standard deviation τ . Right now, the meta analysis focuses on μ and associated inferential statistics. The discussion of heterogeneity describes a set of statistics that are mostly focused on whether the amount of heterogeneity is statistically significant. I'm pretty much uninterested in whether the heterogeneity is statistically significant, though I can imagine that some people are interested in it. For me, however, I think the τ parameter itself -- exactly the measure of how dispersed effects are -- describes heterogeneity best. Please report estimates of τ and interpret them for the reader.

We agree that inclusion of τ (T) improves the paper. We have also gone one further and added in T^2 to any main or sensitivity analyses sections that report heterogeneity so that the reader has

access to all standard measures of heterogeneity (p.15, line.11; p.15, line 22; p.16, line 7; p.16, line 17)

Smaller comments

1. I was unfamiliar with Egger's regression and trim-and-fill, so I had to look them up. Thank you for the chance to learn about them! I think a sentence or two describing the purpose of these procedures and what problem they try to solve would help readers who like me were unfamiliar with them.

Thank you and agreed. We have now updated our method sections to include the following (p.11):

“Funnel plots and Egger’s regression were used to detect small study bias (funnel plot asymmetry), in which smaller studies have larger effect sizes. This can indicate publication bias or other forms of bias. Where these funnel plots suggested that bias was present, the trim and fill method was used to produce the best estimate of the unbiased effect size [55, 58]. This approach identifies any potential funnel plot asymmetry and imputes “missing” studies which should remove the asymmetry.”

2. Style comments: (a) "There are many problems currently facing nations around the world." is a weak opening sentence. (b) I doubt this primacy claim: "The current study aims to provide the first systematic synthesis of the evidence for changing public attitudes and support for policies by communicating evidence about a policy's effectiveness at achieving its goal." (c) lots of extra parentheticals and i.e.,'s and e.g.,'s that could be smoothed out.

Thank you and agreed:

- a) We have removed the opening sentence.
- b) We have altered the wording here:

“The current study aims to systematically synthesise the evidence for changing public attitudes and support for policies by communicating evidence about a policy’s effectiveness at achieving its goal. This is the first study of which we are aware to conduct such a synthesis.”

- c) We have made several changes to remove superfluous parentheticals and the use of “i.e.” and “e.g.”.

3. When a study has two outcome variables and the mean effect is used, how do you calculate the standard error / confidence interval? The two estimates certainly aren't independent, and we can't just take the mean of the standard errors/ confidence intervals. Were these calculated from the raw data and not from summary statistics?

Pooled means were used, following the guidance in Borenstein p.226-7 2009.

We used summary statistics that we extracted from the included studies, with independence assumed as individual raw data were not available. The resulting variance estimates are more conservative (i.e. larger) than those that would have been produced if we used estimates that considered the correlation (i.e. covariance). This conservative estimate of variance would therefore not have increased the possibility of type 1 errors occurring.

The confidence intervals were estimated by the metafor package in R as part of the mixed effects model. (Viechtbauer, W. 2010 Conducting meta-analyses in R with the metafor package. *Journal of Statistical Software*. 36, 1-48; <https://www.jstatsoft.org/article/view/v036i03>)

4. "the intervention that offered the least confounded version of a treatment was selected (e.g., an intervention containing evidence of effectiveness would be chosen over an intervention containing evidence of effectiveness and information on the nature of the problem)" Because the word confounded has a specific statistical meaning that is not what is meant here, I would suggest the authors reword.

Thank you. We have removed this part. The updated section now reads:

“when multiple interventions were eligible, the intervention containing evidence of effectiveness alone would be chosen over an intervention containing evidence of effectiveness *and* information on the nature of the problem.”

5. This sentence inappropriately slips into causal language: "none of the moderators significantly affected the size of the effects." Predicts? Associated? [I'd also mention the low power of such tests for plausibly-sized differences in effect size]

Good point. We have amended this:

“As seen in Table 1, there was no evidence that policy domain, presentation of effectiveness, readability, or the presence of uncertainty moderated the size of the effects”

6. I like that I got to look at the code and data. Thank you!

You are welcome!

#####

Reviewer: 2

Comments to the Author(s)

I think this is a neatly written paper, and as it is a first systematic contribution to a relevant research question, it has merits.

I have, however, some concerns which I could not dismiss by reading the paper. These might be due to my insufficient knowledge of meta-analytic methods, but it would be appreciated if the authors could address them in a revised version or, if they are not relevant, by simply saying so in their reply and explaining why.

1) It is somehow surprising that all eligible studies are randomized experiments. If this is true, it is an interesting result by itself... No one has ever explored ex-post, with real data, how an information campaign on the effectiveness of a policy has influenced public support? How confident are the authors that only hypothetical survey with experimental information provision exist?

Thank you for this comment and we understand the potential concern. However, our search strategy was developed using the Cochrane handbook, with feedback from an information scientist, and we searched a large number of databases, more than are typically searched in systematic reviews. We also screened potentially eligible papers in duplicate, as is recommended, to reduce the likelihood that we would miss a relevant paper. Of course, it is always possible that a paper is missed, but we took all recommended steps to minimise this possibility.

There have been some studies, as you suggest, where public support for policies is monitored before and after a policy is implemented. These provide interesting data but were ineligible for our review which aimed to isolate the effect of messages that contained information on policy effectiveness. In real-world examples, any number of different messages might be disseminated from a number of different sources.

2) The assumption is that participants rely on the information on effectiveness provided within the study, but they may be aware of other/conflictual information. How this is controlled for in the various studies?

You are correct: we are making an assumption that it is the information on effectiveness that drives the results. We believe that this is a reasonable – and standard – assumption that lies at the heart of the scientific method. In relation to the current review we would make the following points regarding the validity of our assumption:

First, included studies comprised randomised experiments, the only difference between control and intervention groups (assuming balanced confounding variables) is the intervention *i.e.* information about policy effectiveness.

Second, given that some interventions contained messages about things other than evidence of policy effectiveness, we conducted a follow-up analysis that included studies that only communicated evidence of effectiveness. The effect sizes were nearly identical leading us to conclude that evidence of effectiveness drives the main change in policy support.

Third, in the few studies that assessed the hypothesised mediating belief - that the policy is effective - the interventions changed this belief. One study conducted a formal mediation test which confirmed this mediation.

We think that these three key observations should help mitigate concerns about this assumption.

3) While not being an expert, I am aware that meta-analysis is 'acceptable' with as few as two studies. However, I struggle with this concept, especially if one wants to control for study heterogeneity (e.g. through random effects). While there is obviously value in the exercise, I wonder to what extent a limited number of studies (35) is credible to generalize the findings of this research. Even if this might be trivial for meta-analysis specialists, a more explicit discussion on why this small number of (all experimental) studies produces generalisable findings would be appreciated, together with a discussion of the limitations. Furthermore, these 35 studies are from 23 different authors if I have computed correctly (one author appears five times, another three times, and two papers are from the lead author of the current study).

We would tend to agree that two studies – while technical possible to enter into meta-analysis – would not provide sufficient evidence to generalise the results but is a principled approach to estimating an average effect. However, we do not think that 35 is a small number for a typical meta-analysis. Within these 35 studies, the sample sizes are large and often nationally representative (across all studies, total $N = 30,858$). Whereas the average Cochrane review contains 6 trials and a total of 945 participants; <https://www.ncbi.nlm.nih.gov/pubmed/12602082>. The primary research in our paper also covers a range of policies - environment, health, immigration, education - and a variety of methods of evidence communication - visual, quantified statements, unquantified statements, narratives, audio. The research was also conducted in many countries. However, for each of these categories, particularly policy and country there was bias. Our results are more influenced by those studies conducted in the USA and in health policy domains than by others. We have now added this limitation to the discussion on p.20:

“The majority of the primary research included in this review was conducted in the USA (25/36) and the majority of policies studied were in the domain of health (20/36). The results of the review may therefore be less generalisable to communicating evidence of effectiveness for policies other than health in countries other than the USA. However, in the meta-regression we did not find evidence that policy domain moderated the effect of communicating evidence but we note that this analysis had too low power to fully exclude this possibility”.

4) Why aren't funnel plots displayed in the paper? A graph of estimates and precision of the included studies would be very informative.

Agreed, thank you. We have moved the funnel plots from the supplemental section to the main manuscript.

5) Not unrelated to the points (3) and (4) is the risk of publication bias. This is mentioned by the authors, and I think that section is valuable... but the discussion is too short and "implicit" to be understood. And how can one elicit publication bias from the funnel plots? How many journals publish negative or null findings from experiments?

We have now elaborated on this point in the discussion:

“There was some evidence of funnel plot asymmetry for the evidence of effectiveness meta-analysis but not for the evidence of ineffectiveness meta-analysis. This may indicate publication bias but other sources of bias could also explain this such as language bias and outcome reporting bias [58]. We addressed this by using the “trim and fill” approach. While this does not eradicate the problem it can reduce it.”

6) "Conversion from the pooled effect sizes to changes in policy support proportions were calculated using a number needed to treat formula (see Table 2; [61])". What does this mean? Can the author be more explicit about their data processing?

Yes, absolutely. The number needed to treat (NNT) formula is often used in epidemiology to work out “how many people do I need to treat to save one person?”. This can then be expressed as follows: “if 100 people receive the intervention (evidence of effectiveness in the current context), then how many people are saved (attitudes changed in the current context)?”. The information you

need to work this out are the baseline rate (in our case, baseline number of people who support a policy), the effect size (which we got from the meta-analysis), and the NNT formula (which we got from the cited paper; <https://doi.org/10.1371/journal.pone.0019070>).

The NNT formula that we used is:

$$NNT = \frac{1}{\Phi(\delta + \Psi(CER)) - CER}$$

In which:

Φ = cumulative distribution function of the standard normal distribution

Ψ = the inverse of Φ

δ = cohen's d

CER = the baseline level of support for a policy

The R code that we used to estimate changes in policy support proportions is available at the very bottom of our "initial meta analysis" supplement.

We have highlighted this last point in the main text (p.13:

"The R code used to estimate changes in policy support proportions can be found in the supplement."

7) I wonder how standard errors are treated in this meta analysis. The authors report large Ns by considering the sample size of the analysed studies, but the number of studies is small. I am not sure that allowing for random effects does the full job. Is there some form of clustering for the standard errors? I would also be curious to see the results from a fixed effect model as a robustness check, I don't see particular reasons why the results should be different?

Agreed. The raw data for the meta-analysis comes from means and standard deviations which we extracted from the included studies. As part of the analysis, these means and standard deviations are converted into a standardised mean difference, and the confidence intervals are created using the variance associated with the effect size.

Fixed effect models were ran as the reviewer requested and the results are very similar to the random effect models that we reported in the paper for the effectiveness analysis:

Random effects model: SMD = .11, 95% CI [.07, .15], $p < .001$

Fixed effects model: SMD = .10, 95% CI [.07, .12], $p < .001$

And the ineffectiveness analysis:

Random effects model: SMD = -.14, 95% CI [-.22, -.06], $p < .001$

Fixed effects model: SMD = -.15, 95% CI [-.20, -.09], $p < .001$

As the reviewer predicted, there is very little difference. We now report these results in the supplement as a robustness check as suggested by the reviewer.

8) Table 2 is extremely relevant and impressive in terms of its 'linearity' and precision... but it's not clear to me how it was estimated, given the extremely low number of studies. I am especially surprised by the relatively narrow confidence intervals. Was a linear effect assumed? See also my comment on clustering above.

We think there may have been a misunderstanding here. These results were not produced from a model so linearity or similar other characteristics of the data were not assumed.

Instead we estimated these values using the same number needed to treat (NNT) formula as we used before and described above. One of the values required in the NNT formula is the baseline rate which in the current context is the baseline level of support. To interpret our main results such as in the abstract, we selected the baseline rate of 50%. While this is a common level of support for a number of policies, it could be misleading if it is the only rate given. We therefore estimated the effect size with baseline rates of support ranging from 10% to 90% which illustrates marginally different results depending on baseline level of support.

RE the narrow confidence intervals – these are the same confidence intervals that were calculated in the main results. They are just fed into the formula and re-expressed on a scale from 0-100%. Using the code we provide in the supplement, it is possible to estimate this for any baseline level of support.

This method is discussed at the end of the synthesis of the Results section, within the method, p.13. We have reworded this to clarify that the confidence intervals are also derived using this method:

“The pooled effect sizes and associated confidence intervals were converted to changes in policy support proportions using a number needed to treat formula (see Table 2; [61]).”

9) (Minor) Page 19 - confectionery instead of confectionary?

Thanks for pointing this out. This has been changed.

Appendix B

We thank the editor and the reviewers for their detailed reading and feedback over multiple versions of this manuscript. We are glad that reviewer 2 recommends acceptance; and following our clarifications and responses to reviewer 1 below we hope that the editor can now accept the paper for publication in RSOS.

Reviewer comments to Author:

Reviewer: 2

Comments to the Author(s)

Thanks for addressing my comments in a neat and accurate way. I have no further concerns, this seems a relevant contribution to the literature.

Reviewer: 1

Comments to the Author(s)

I am grateful to the authors to for their response to my point (2) about the choice of statistical model. I also agree with the changes they made in response to my smaller comments. The paper is now in my view very nearly ready for publication, though I must say I am not convinced by the authors' rebuttal to my critique (1).

The main sticking point appears to be about bias due to chance imbalances that crop up in some randomizations. The authors describe studies as "weak" if they do not discuss covariate adjustment, or matching, or other adjustment for pre-treatment covariates. In their response memo, the authors justify this choice by pointing to references that describe how when covariates are out of balance, estimates tend to be further from the truth. That claim is true with respect to a particular estimate, but it is NOT true with respect to an estimator. The estimators are unbiased regardless of covariate adjustment, though in the absence of adjustment, estimates can be far from the estimand.

Indeed, the **main point** of a randomized experiment is to provide an unbiased procedure that gets the answer right on average. Bias is defined as a property of the procedure over the set of possible randomizations, NOT as a property of the distance between the estimate and the estimand in any single realization. When we do covariate adjustment, it's not to correct bias, it's to reduce variance, i.e., by systematically drawing estimates that are far from the truth in towards the truth. This is not about bias -- both the unadjusted and adjusted procedures are (approximately) unbiased for the ATE. This paper should emphatically not claim that "not reporting whether groups were matched for confounding variables was also a large factor". In a randomized trial, this consideration is about precision, not bias. I care about this point, and I would encourage the authors to please reconsider their position.

I also do not think that it is appropriate to grade studies as weak because they didn't use a measurement scale that had been specifically validated. Those that have may still be quite weak, and those that haven't may be quite

strong. This quality discussion, I think, is misplaced. I think the paper would be improved by removing the "Risk of Bias within studies" altogether.

We thank the reviewer for their in depth feedback on this section. We have discussed this as a team and come to a decision which allows us to keep the quality assessment while taking on board the reviewer's comments.

Firstly, to reiterate our general position on this topic: it is very important that quality assessment of studies is done when conducting a systematic review. One of the PRISMA guidelines (<https://annals.org/aim/article-abstract/744664>) for reporting and conducting systematic reviews is as follows:

Risk of bias in individual studies

12 Describe methods used for assessing risk of bias of individual studies (including specification of whether this was done at the study or outcome level), and how this information is to be used in any data synthesis.

As we discussed in our last response, different tools are appropriate for different domains, and we chose the one that we judged to be most appropriate for this topic area.

That being said, we agree with the reviewer that that purpose of using covariates is to improve precision of estimates, not try to reduce bias and we have therefore removed this variable from the overall quality coding score. We did not take this decision lightly, as we aimed to follow best practice by pre-registering our methods. Any changes to the methods after running the analysis introduces bias and therefore we have highlighted this as a deviation from our protocol in the main document.

"One category – confounders – was not factored into the quality score following advice from a reviewer." (p10-11)

There are also several consequences to this decision:

1. This changes the overall quality score for a number of the papers in the review (fewer are scored as weak, more are scored as moderate)

"For the evidence of effectiveness meta-analysis, 26 studies were rated as weak (high risk of bias), nine were rated as moderate, and none were rated as strong (low risk of bias)" (p.14)

2. As more studies are now classified as "moderate", more are eligible for the sensitivity analysis, which we have now run. The results appear to strengthen our primary conclusions

"Excluding studies at high risk of bias resulted in $k = 9$ effect sizes and $N = 12,527$. Communicating evidence that a policy was effective increased support for the policy, $SMD = .12$, 95% CI [.04, .20], $p = .002$. There was substantial and significant heterogeneity, $Q (8) = 27.06$, $p < .001$, $I^2 = 65\%$, $T = .09$, $T^2 = .01$." (p.15)

3. As some studies within the ineffectiveness meta-analysis are now moderate, we have run a sensitivity analysis on those. The analysis was not statistically significant – but this isn't surprising given the small N and k . However, the effect size falls within the 95% confidence intervals from the primary analysis, suggesting that the main ineffectiveness meta-analysis result could be robust

“Sensitivity analysis. *The main analysis was re-run to test whether the significant overall effect remained after excluding the studies that were at high risk of bias. Excluding studies at high risk of bias resulted in $k = 3$ effect sizes and $N = 1198$. There was no evidence that communicating evidence of ineffectiveness on policy changed support for policies, $SMD = -.08$, 95% CI $[-.20, .03]$, $p = .155$. There was no evidence of heterogeneity, $Q(2) = 0.87$, $p = .648$, $I^2 = 0\%$, $T = .00$, $T^2 = .00$.”* (p.16)

4. We have changed our discussion section accordingly and removed the parts referring to the balancing of covariates as the reviewer recommended

“The quality assessment tool assesses a range of different factors. In particular, one of these factors received lower quality ratings across studies: outcome validity and reliability. Only one study reported tests that their outcome measure was valid [67].” (p.19)

One point that the reviewer brought up is the validity and reliability checking of primary outcome. It should be noted that if a study is scored as weak overall, this is not simply due to a “weak” score on this criteria, but for multiple “weak” scores across categories. We believe that ensuring that variables are valid and reliable is an important part of research. Study quality checks have an additional benefit: they can help direct future research. By highlighting that the primary outcomes in this field tend not to have received appropriate validity checking, we hope that others will see this as an opportunity to improve in the future. We have stated this at the end of this discussion section on study quality:

“Future research would benefit from improving reporting standards, including confirming the validity and reliability of policy support measures” (p.19)

One final point. The authors write in their rebuttal that the variance estimates for pooled are more conservative than if the authors had used estimates that included the correlation across studies. Is this a worst-case variance estimator? If so, I think the text needs more detail on that procedure. Most naive approaches I can think of would not necessarily be conservative. In any case, I can't tell from the text what that variance estimator is, so I can't evaluate the claim that it is conservative. Some clarification on this would be appreciated.

Apologies for any confusion. We did assume a worst case scenario. We have added in an extra sentence pointing readers to the formulae and I have typed them up below to save you finding the book (Borenstein, M., Hedges, L. V., Higgins, J. P., Rothstein, H. R. 2011 *Introduction to meta-analysis*. West Sussex, UK: John Wiley & Sons; p.226-7).

Our updated methods section (p10-11):

“in studies that included multiple eligible outcome measures, the combined means and variances were calculated using standard formulae (equations 24.1 & 24.2 [55])”

These formulae are, for means:

$$\bar{Y} = \frac{1}{2}(Y_1 + Y_2)$$

And for variances:

$$V_{\bar{Y}} = \frac{1}{4}(V_{Y_1} + V_{Y_2} + 2r\sqrt{V_{Y_1}}\sqrt{V_{Y_2}})$$

Those remaining sticking points notwithstanding, I think this is a very nice meta analysis that I'll be citing in the future. My thanks to the authors.